# Differentiation of Human Cardiac Atrial Appendage Stem Cells into Adult Cardiomyocytes: A Role for the Wnt Pathway?

**DOI:** 10.3390/ijms21113931

**Published:** 2020-05-30

**Authors:** Leen Willems, Annick Daniëls, Yanick Fanton, Loes Linsen, Lize Evens, Virginie Bito, Jeroen Declercq, Jean-Luc Rummens, Karen Hensen, Marc Hendrikx

**Affiliations:** 1UHasselt, Faculty of Medicine and Life Sciences, Martelarenlaan 42, 3500 Hasselt, Belgium; leen.willems@uhasselt.be (L.W.); yanick.fanton@beta-cell.com (Y.F.); loes.linsen@uzleuven.be (L.L.); lize.evens@uhasselt.be (L.E.); virginie.bito@uhasselt.be (V.B.); jeroen.declercq@uzleuven.be (J.D.); jean-luc.rummens@jessazh.be (J.-L.R.); karen.hensen@medtronic.com (K.H.); 2Laboratory of Experimental Hematology, Jessa Hospital, Stadsomvaart 11, 3500 Hasselt, Belgium; annick.daniels@jessazh.be; 3University Biobank Limburg, Jessa Hospital, 3500 Hasselt, Belgium

**Keywords:** cardiac progenitor cells, cardiac regeneration, differentiation, proliferation, Wnt pathway

## Abstract

Human cardiac stem cells isolated from atrial appendages based on aldehyde dehydrogenase activity (CASCs) can be expanded in vitro and differentiate into mature cardiomyocytes. In this study, we assess whether Wnt activation stimulates human CASC proliferation, whereas Wnt inhibition induces cardiac maturation. CASCs were cultured as described before. Conventional PCR confirmed the presence of the Frizzled receptors. Small-molecule inhibitors (IWP2, C59, XAV939, and IWR1-endo) and activator (CHIR99021) of the Wnt/β -catenin signaling pathway were applied, and the effect on β-catenin and target genes for proliferation and differentiation was assessed by Western blot and RT-qPCR. CASCs express multiple early cardiac differentiation markers and are committed toward myocardial differentiation. They express several Frizzled receptors, suggesting a role for Wnt signaling in clonogenicity, proliferation, and differentiation. Wnt activation increases total and active β-catenin levels. However, this does not affect CASC proliferation or clonogenicity. Wnt inhibition upregulated early cardiac markers but could not induce mature myocardial differentiation. When CASCs are committed toward myocardial differentiation, the Wnt pathway is active and can be modulated. However, despite its role in cardiogenesis and myocardial differentiation of pluripotent stem-cell populations, our data indicate that Wnt signaling has limited effects on CASC clonogenicity, proliferation, and differentiation.

## 1. Introduction

Heart failure after myocardial infarction (MI) is the leading cause of mortality worldwide and results from the loss of cardiomyocytes and remodeling of the remote cardiac tissue due to coronary occlusion. While heart transplantation is currently the only available therapy to fully restore cardiac functionality after an MI, its major drawbacks, such as life-long immunosuppressive therapy and the paucity of donor hearts, do not allow it to be a realistic option for every patient [1]. In that context, the development of cardiac regeneration strategies using various stem-cell types including pluripotent stem cells (PSCs), skeletal myoblasts, bone-marrow derived cells (BMCs), and endogenous cardiac stem cells (CSCs) is raising a lot of interest. However, up to now, the results obtained in different clinical trials were rather disappointing because of the chosen cell type for transplantation and its characteristics upon differentiation [2].

Resident CSCs might offer better promise for cardiac regeneration as they are most likely pre-programmed to become cardiomyocytes. Several approaches were used to isolate and identify endogenous cardiac progenitor populations, including c-kit, Sca-1, aldehyde dehydrogenase (ALDH), side population, and cardiospheres [3]. The c-kit^+^ CSCs are the most studied, but recently also the most challenged CSC type. Beltrami et al. were the first to describe a c-kit^+^ CSC population in rat hearts which gave rise to cardiomyocytes, smooth muscle cells, and endothelial cells, and which improved cardiac function after injection in a rat MI model [4]. This CSC type was also identified later in humans where they displayed the typical characteristics of stem cells [5]. Although other studies were not able to replicate these results, c-kit^+^ CSCs were used in the Stem Cell Infusion in Patients with Ischemic Cardiomyopathy (SCIPIO) phase I clinical trial [6]. However, concerns were raised on the integrity of those data and the paper was retracted [7]. Meanwhile, three papers in Circulation and 10 papers in Circulation Research, forming the backbone of the findings on c-kit cells were retracted because of fraudulent data. Furthermore, van Berlo et al. showed that 0.008% of the cardiomyocytes in the adult heart are derived from c-kit^+^ cells and these cells only minimally contributed to cardiac regeneration after MI in a mouse model [8]. These results were confirmed by Sultana et al. [9]. 

Altogether, data indicate that endogenous c-kit^+^ CSCs are most likely not the primary cell source for cardiomyocyte renewal in adults and might, therefore, not be the appropriate CSC type for future cardiac regeneration therapies. The value and the importance of c-kit^+^ CSCs cells to regenerate cardiac tissue are, therefore, questionable.

In contrast, cardiac atrial appendage stem cells (CASCs) are typically ALDH bright (ALDH^br^), cluster of differentiation 34 (CD34)^+^, Isl1^+^, CD45^−^ and c-kit^−^, although CD34 expression is lost during cell culture [10]. A similar ALDH^br^ CSC population was also described in mice [11]. In addition, CASCs express multiple pluripotency associated genes, including *OCT-4, NANOG, C-myc, KLF4, lin-28, DPPA3*, and *Tbx3*, and they possess a clonogenicity of around 20%, confirming a stem-cell phenotype. Co-culturing CASCs with NRCMs resulted in cardiomyocyte differentiation with a sarcomeric organization of cardiac troponin T (cTnT) and I (cTnI) [10]. Their potential was further confirmed in vivo in a Göttingen minipig model of acute MI where autologous CASC transplantation preserved global and regional left-ventricular function due to extensive engraftment and myocardial differentiation of the transplanted cells [12]. In contrast to the extensive myocardial differentiation potential, CASCs show limited differentiation toward endothelial cells, but stimulate angiogenesis via paracrine mechanisms [12,13]. Indeed, the replacement of lost heart muscle by new functional cardiomyocytes, as well as new blood vessel formation, is essential to ensure full cardiac repair.

The molecular and cellular mechanisms involved in CASC proliferation and differentiation remain largely unknown, but understanding these mechanisms could further enhance cardiac regeneration therapies. In that context, the Wnt/β-catenin pathway could be heavily involved since it is essential for vertebrate cardiogenesis and is re-activated in response to cardiac injury by regulating transcription of target genes involved in cell differentiation and proliferation [14,15]. Therefore, we investigated its precise role in CASC proliferation and differentiation which can eventually lead to improved CASC-based therapies.

The canonical Wnt/β-catenin signaling pathway is activated when secreted Wnt ligands bind to the receptor complex, composed of a seven-transmembrane Frizzled (FZD) receptor and a low-density lipoprotein receptor-related protein (LRP)5/6 co-receptor. This interaction leads to the inhibition of glycogen synthase kinase (GSK)-3β and subsequently to a hypophosphorylation of β-catenin. As a result, accumulation of β-catenin in the nucleus activates transcription of specific target genes, including *Cyclin D1*, V-Myc avian myelocytomatosis viral oncogene homolog (*cMYC*), and Jun proto-oncogene (*cJUN*) (Figure 1) [16]. During cardiogenesis, the effects of canonical Wnt signaling occur in four distinct phases. Canonical Wnt signaling is involved in the induction of mesoderm formation, followed by a subsequent inhibition for cardiac specification and cardiac progenitor cell formation. Later on, proliferation of these progenitor cells is positively regulated by canonical Wnt signaling, while inhibition of this pathway is essential for terminal cardiac differentiation [17]. The identification of these distinct phases led to the development of new differentiation protocols for PSCs which promote cardiac differentiation by sequentially activating and inhibiting the canonical Wnt signaling pathway [18]. We hypothesize that the same processes are involved in CASCs. However, in preliminary experiments sequentially activating and inhibiting the Wnt pathway in CASCs, we were unable to induce significant variations of cardiac gene expressions at five and 15 days compared to control. Immunofluorescence for TnT and TnI was negative. We, therefore, aimed to identify whether Wnt activation is able to stimulate CASC proliferation and whether Wnt inhibition would induce cardiac maturation. Since Wnt signaling has distinct roles during the different phases of cardiac development, we also studied the origin of CASCs by determining the localization of CASCs in adult heart and the cardiac differentiation stadium of CASCs in culture.

## 2. Results

### 2.1. CASCs Are Predominantly Present in the RAA

Firstly, the distribution of CASCs in the adult heart was determined by measuring the proportion of ALDH^br^ cells within various compartments of the heart. Within the same patient, the proportion of ALDH^br^ cells was significantly higher in the right atrial appendage (RAA) (5.1% ± 4.4%) compared to the left atrial appendage (LAA) (3.46% ± 2.6%; *p* < 0.05; *n* = 33) (Appendix A). To analyze the distribution of CASCs in other regions of the heart from which human samples are not as easily obtained, the presence of ALDH^br^ cells in various compartments was studied in adult pig hearts. As shown in Table 1, ALDH^br^ cells were predominantly present in LAA and RAA, corresponding to the data obtained from human atrial appendages. ALDH^br^ cells were almost absent in the left ventricle and septum and could be found at low levels in the atria, the right ventricle, and the apex (Appendix A). In general, although there was no significant difference between left and right in pigs due to the small sample size, ALDH^br^ cells appeared to be more abundant in the right than in the left part of the heart.

### 2.2. CASCs Express Early Cardiac Differentiation Markers during Expansion

To identify the cardiac differentiation stadium of human CASCs during expansion, a number of early- and late-stage cardiac specific markers were evaluated in ALDH^br^ cells (Figure 2). As previously described, the ALDH^dim^ population could not be cultured after isolation [19]. For the pre-cardiac mesoderm markers, only kinase insert domain receptor (*KDR*) was expressed in CASCs, as well as in adult atrial and ventricular heart muscle, while Brachyury (*T*) was absent. Furthermore, the early cardiac transcription markers such as GATA binding protein 4 (*GATA4*), T-box 5 (*TBX5*), T-box 18 (*TBX18*), and NK2 homeobox 5 (*NKX2.5*) were expressed in human CASC cultures, as well as in adult heart muscle. From the mature cardiac markers typically found in adult cardiomyocytes, only cardiac type troponin T2 (*TNNT2*) and myosin light chain 2 (*MYL2; MLC-2v*) were expressed in human CASC cultures, although at lower levels as compared to adult atrial and ventricular tissue. Myosin light chain 7 (*MYL7; MLC-2a*) and hyperpolarization-activated cyclic nucleotide-gated potassium channel 4 (*HCN4*) were present in adult heart muscle but were completely absent in CASCs. These data suggest that human CASCs display a cardiac progenitor-like phenotype rather than a mature cardiac phenotype during expansion.

### 2.3. Several FZD Receptor Subtypes are Expressed in CASCs

When expanding CASCs for clinical use, it would be beneficial to reduce the expansion time by stimulating CASC proliferation. This might be achieved by interfering with the canonical Wnt pathway. Since binding of the Wnt ligand to the FZD receptor is essential for the activation of the downstream Wnt/β-catenin pathway, we firstly analyzed the expression pattern of several FZD receptors in CASCs by conventional PCR. As shown in Figure 3, expression of *FZD1*, *2*, *4*, *6*, and *7* was already detected after 25 cycles, indicating abundant expression levels of these FZD subtypes.

### 2.4. Wnt Signaling Can Be Modulated in CASCs by Specific Small-Molecule Activators and Inhibitors

To test if the Wnt/β-catenin pathway could be modulated in CASCs, we investigated whether the levels of total and active β-catenin (dephosphorylated on Ser37 or Thr41) could be modified by CHIR99021 (small-molecule Wnt activator) or C59, IWP2, XAV939, and IWR1-endo (small-molecule Wnt inhibitors).

As shown in Figure 4A, 6 µM CHIR99021 significantly increased the levels of total and active β-catenin two-fold and five-fold in CASCs, respectively (*p* < 0.05). 293T cells, used as a positive control, showed a 23-fold and 26-fold increase in total and active β-catenin levels. As expected, CHIR99021 treatment did not upregulate total or active β-catenin levels in the SW480 cell line, due to an adenomatous polyposis coli (APC) mutation which inhibits β-catenin ubiquitination [20]. Finally, CHIR99021 treatment slightly but significantly reduced cell viability in both CASCs and control cell lines (Figure 4B).

To investigate whether Wnt/βcatenin signaling could be inhibited in CASCs, we applied various small-molecule inhibitors targeting different levels of the Wnt pathway. As shown in Figure 5, 4 µM IWP2 or 1 µM C59, blocking Wnt ligand production and secretion, did not affect total or active β-catenin levels, in both CASCs and SW480 cells. In contrast, treatment with 2 µM XAV939 or 4 µM IWR1-endo, stabilizing the APC/Axin/GSK-3β destruction complex of β-catenin, significantly reduced active β-catenin levels by 52% and 54%, respectively (*p* < 0.05), indicating a potent inhibition of Wnt/β-catenin signaling in CASCs. Total β-catenin levels remained unchanged in CASCs (Figure 5A). In SW480 control cells, total and active β-catenin levels decreased with by 89% and 99%, respectively (*p* < 0.05), after XAV939 treatment and by 91% and 100%, respectively (*p* < 0.05), after IWR1-endo treatment (Figure 5B). Wnt inhibitors, in contrast to CHIR99021, did not affect cell viability (data not shown).

### 2.5. Modulating Wnt Signaling Does Not Influence CASC Clonogenicity or Proliferation

Given that β-catenin levels could be modulated in CASCs, we investigated whether Wnt stimulation could influence clonogenicity and proliferation of CASCs in a serial cloning experiment. As shown in Figure 6, CHIR99021 treatment had no effect on the number or the size of primary, secondary, or tertiary clones suggesting that Wnt activation has little or no effect on CASC clonogenicity or proliferation. Furthermore, tertiary clones were not large enough to replate them for quaternary cloning, indicating that the cloning activity of CASCs in both conditions was exhausted at that time point.

To further analyze the functional effect, we evaluated the expression levels of three proliferation-associated Wnt target genes, *cMYC*, *cJUN*, and *Cyclin D1* (Figure 7). Activating Wnt signaling did not modify the messenger RNA (mRNA) levels of *Cyclin D1* and *cJUN* in CASCs. In 293T cells, CHIR99021 treatment induced a near two-fold (*p* < 0.05) increase in *Cyclin D1* and *cJUN* expression (Figure 7A,B, left panels). When evaluating the effect of Wnt inhibition on proliferation associated genes, we found that none of the inhibitors tested changed the expression levels of *Cyclin D1* in CASCs (Figure 7A right panel), and *cJUN* expression in CASCs only decreased by 25% (*p* < 0.05) upon XAV939 treatment (Figure 7B, right panel). In SW480 cells, there was a decrease by 35% and 32% (*p* = 0.0188) in *Cyclin D1* expression after XAV939 and IWR1-endo treatment, respectively (Figure 7B right panel). Furthermore, *cMYC* expression decreased by 39% (*p* < 0.05) in SW480 cells after IWR1-endo treatment (Figure 7C right panel). Both Wnt stimulation and inhibition reduced *cMYC* levels in CASCs, by 48% (*p* < 0.05) after CHIR99021 treatment (Figure 7C left panel) and by 35% (*p* < 0.05) after XAV939 treatment (Figure 7C right panel). Our data indicate that Wnt inhibition or activation has no or little effect on the expression of proliferation-associated genes in CASCs, confirming the results of the serial cloning experiment.

### 2.6. Wnt Inhibition in CASCs Does Not Induce Mature Cardiomyocyte Differentiation

Finally, a potential effect of Wnt inhibition on CASC differentiation was investigated. To this end, CASCs were treated with 2 µM XAV939 or 4 µM IWR1-endo, which effectively inhibited canonical Wnt signaling as shown above. Subsequently, changes in mRNA expression of different cardiac differentiation markers (*GATA4*, *NKX2.5*, *TBX5*, *TBX18*, *TNNT2*, and *MYL2*) were examined. As shown in Figure 8, *GATA4* expression increased after XAV939 treatment, but this could not be reproduced with IWR-1endo treatment. IWR1-endo significantly increased *TBX5* expression and decreased *TNNT2* expression, but this could not be reproduced with XAV939 treatment. *NKX2.5*, *TBX18*, and *MYL2* expression in CASCs was not affected by Wnt inhibition. These data demonstrate that Wnt inhibition alone is insufficient to induce mature cardiomyocyte differentiation in CASCs.

## 3. Discussion

A lot of questions related to cell-based cardiac regeneration therapies remain unresolved. For example, which is the ideal cell type to transplant? What is its origin? Which is the optimal expansion method? Do they need to be pre-differentiated or in any way treated before administration? All these issues will need to be addressed in an attempt to improve current stem-cell strategies and to make way for their successful clinical translation. In this study, we show that CASCs are predominantly present in atrial appendages and more so in the right than in the left. Furthermore, studying the cardiac differentiation stadium of CASCs during expansion suggests that they are an intermediate already committed toward myocardial differentiation. Finally, despite the presence of canonical Wnt signaling and unlike other stem-cell populations, modulating Wnt signaling has a limited effect on CASC proliferation or differentiation.

In the last decade, a lot of effort was put into the development of cardiac regeneration therapies for heart failure patients. Although previous clinical studies indicated that BMC- and CSC-based therapies can be safely performed, functional outcomes were rather disappointing [2]. Therefore, selecting the most appropriate stem cell type and understanding the mechanisms driving its proliferation and differentiation are fundamental. Previous studies already suggested CASCs as good candidates for cardiac regeneration therapies [10,12]. So far, CASCs are routinely isolated from pig and human atrial appendages. However, whether this location is unique or whether they can also be found in other parts of the heart remained largely unknown. In this study, we showed that CASCs are predominantly present in atrial appendages and more so in the right than in the left, both in humans and in pigs. This precludes harvest via endomyocardial biopsies, but requires a surgical approach. In addition to a conventional approach via sternotomy in the context of heart surgery, the atrial appendages are also relatively easy and safe to obtain via minimal (port-)access surgery. Moreover, CASCs are more abundant in atria than in ventricles. Similar findings for ALDH^br^ cells in eight-week old mouse hearts were obtained by Roehrich et al. [11].

When studying the cardiac differentiation stadium of CASCs during expansion, qPCR analysis demonstrated expression of *KDR*, *NKX2.5*, *GATA4*, *TBX5*, *TBX18*, *TNNT2*, and *MYL2*, but not of *T*, *MYL7*, and *HCN4*. *KDR* and *T* are both markers for early mesoderm and show similar expression levels for CASCs and adult heart. *TBX18* expression might confirm a possible pro-epicardial origin of CASCs [21]. However, the early cardiac-specific transcription factors *NKX2.5* and *GATA4,* known to be expressed in first and second heart field (FHF and SHF) progenitors, are also expressed in CASCs. Recently, early expression of *HCN4* was also reported as a marker for FHF progenitors [22]. However, this marker was not expressed in CASCs, while another FHF marker *TBX5* was [23]. These results suggest a possible heterogeneous embryonic origin of CASCs, as also suspected for other CSC types [24]. However, elaborated transcriptomics studies or lineage tracing experiments are needed to further unravel the precise origin and nature of CASCs. Although the expression levels of the different markers were very similar amongst individual patient’s CASC cultures, a broader screening in a larger patient population is necessary to identify inter-patient differences and whether this can be linked to differences in CASC proliferation, integration, and differentiation, as well as to differences in functional outcomes. To further investigate the heterogenic character of CASCs, single-cell analyses, such as single-cell sequencing, might also be useful to reveal cellular differences within a patient-specific CASC population.

The expression of *NKX2.5*, although lower than in the atrium and ventricle, might suggest a limited endothelial differentiation potential of CASCs. Wu et al. demonstrated that *NKX2.5* expression comprises endothelial differentiation in cardiac-specific *NKX2.5*-positive cells from the developing mouse embryo [25]. This was confirmed by the limited endothelial cell differentiation of less than 1% in the Göttingen minipig model for acute MI [12]. Despite this limited endothelial differentiation capacity, we recently demonstrated that CASCs stimulate angiogenesis via paracrine mechanisms through the secretion of numerous growth factors that promote important steps of new blood vessel formation [13]. The expression of *TNNT2* and *MYL2* in CASCs suggests that CASCs can be considered as an intermediate already committed toward myocardial differentiation. The profound myocardial differentiation potential was already confirmed both in vitro and in vivo. Indeed, co-culturing CASCs with neonatal rat cardiomyocytes induced sarcomeric organization of cTnT and cTnI while transplanting CASCs in the left ventricle of a Göttingen minipig model for acute MI-induced sarcomere formation and protein expression of cTnT, cTnI, connexin 43 (CX43), and MLC-2v [10,12].

Although CASCs can efficiently differentiate into cardiomyocytes, we observe a high variability between patients with respect to their proliferation potential. In addition, it is known that myocardial differentiation is only obtained when CASCs are in close contact with functional cardiomyocytes [10,12,19]. Therefore, more insight into the molecular mechanisms involved in CASC proliferation and differentiation might be useful to amplify the proliferation potential of CASCs and to understand the essential cues that drive CASCs toward myocardial differentiation. Several factors, such as Wnt, FGF, BMPs, Notch, and Hedgehog are known to influence cardiac progenitor cell proliferation and differentiation [26]. In this study, we focused on the role of canonical Wnt signaling in adult cardiac progenitor cells, more specifically its effect on CASC proliferation and differentiation. Firstly, we showed that CASCs express FZD receptors *FZD1*, *2*, *4*, *6*, and *7*. According to human gene database GeneCards, these specific FZD receptors are also known to be expressed in adult heart. Furthermore, CHIR99021 treatment upregulated total and active β catenin, while XAV939 and IWR1-endo reduced active β-catenin expression. However, CHIR99021 treatment resulted in a slight but significant decrease in cell viability. This is most likely caused by a toxic effect of the product and not the result of Wnt stimulation, since cell toxicity was also observed in SW480 cells where Wnt signaling was not further upregulated. This cytotoxic effect of CHIR99021 was also observed in mouse embryonic stem cells [27]. Since CHIR99021 treatment did not increase the number or the size of the formed clones, Wnt activation had no or limited effect on CASC clonogenicity or proliferation. Furthermore, no upregulation in the proliferation-associated genes *Cyclin D1*, *cMYC*, and *cJUN* was detected in CASCs after Wnt activation. Instead, we observed a decrease in *cMYC* expression levels, both in CASCs and in 293T cells. However, a similar decrease in *cMYC* expression was also seen after inhibiting canonical Wnt signaling in CASCs and SW480 cells. The reason why both Wnt upregulation and downregulation induce a decrease in *cMYC* expression is unknown, but a negative effect of *cMYC* on cardiac function by inducing cardiomyocyte hypertrophy was already described [28]. Furthermore, Wnt activation seems to differently affect cell proliferation, depending on the CSC type [29,30]. Indeed, no effect of Wnt activation was observed on CASC proliferation, while it is known to enhance embryonic and postnatal Isl1-positive cardiovascular progenitor cell proliferation [29] but decrease adult cardiac side population cell proliferation [30].

Finally, the effect of Wnt inhibition on CASC differentiation was examined. Current differentiation protocols for PSCs use a sequential incubation with small-molecule Wnt activators and inhibitors to drive cardiomyocyte differentiation [18]. In CASCs, Wnt inhibition only upregulated the early cardiac differentiation markers *GATA4* and *TBX5*, while downregulating the mature cardiac differentiation marker *TNNT2*. The upregulation of *GATA4* after Wnt inhibition is in line with the findings of Afouda et al., showing that canonical Wnt signaling inhibits *GATA4* expression and that this restricts cardiogenesis in *Xenopus* animal cap and cardiac mesoderm explant assays [31]. A study of Zelarayan et al. also showed an increase in *GATA4* and *TBX5* expression after β-catenin depletion in mouse Sca-1-positive cardiac progenitor cells. However, as opposed to our study, this was also accompanied by an increase in the number of cTnT-positive cells [32]. This difference is likely to be species- and cell-type-related. Furthermore, Foley and Mercola showed that inhibition of the canonical Wnt pathway induces the expression the early cardiac differentiation genes *TBX5* and *NkX2.5*, but not the mature myocardial differentiation markers cTnI and myosin heavy chain (MHC)α [33]. Finally, Burridge et al. also showed a decrease in cTnT-positive cells after treatment with the Wnt inhibitor XAV939 during the early mesoderm differentiation of PSCs [34]. These findings suggest that Wnt inhibition upregulates the early cardiac markers but hinders differentiation toward mature cardiomyocytes.

In this study, cells were incubated for 48 h with these various treatments, after which gene expression was *immediately* analyzed. However, we have strong evidence that the obtained results would not have been different had the cells been analyzed at a later point in time.

While developing an “in vitro” potency test for cardiomyogenic differentiation, CASCs (50.000 cells/well) were seeded in mesenchymal stem cell (MSC)-brew good manufacturing practice (GMP) medium in six-well plates. After 24 h, the medium was replaced by a differentiation medium (Roswell Park Memorial Institute (RPMI) 1640, 213 µg/mL l-ascorbic acid 2-phosphate, and 500 µg/mL rice-derived recombinant human albumin), with the Wnt-inhibitor IWP-2 for two days, and then replaced by the differentiation media alone for the remainder of the process. Cells were analyzed after five and 15 days in media alone for cardiomyogenic-specific gene expression, including *Nkx2.5, MEF2C, TNNT2*, and cardiac *actin*. This protocol was unable to induce significant variations of cardiac gene expressions at five and 15 days as compared to control, indirectly indicating that analyzing the β-catenin activation/inhibition or gene expression at a later stage after Wnt inhibition would not have altered the result (data not shown).

## 4. Materials and Methods

### 4.1. Patients

All procedures were carried out in accordance with the principles set forth in the Helsinki Declaration. Approval by the Jessa Institutional Review Board and informed consent from each patient were obtained. Human CASCs were isolated from atrial appendages obtained from ischemic heart disease patients undergoing routine cardiac surgery, as previously described by Koninckx et al. [35]. In brief, tissue samples were minced and enzymatically dissociated with collagenase II (600 U/mL; Invitrogen, Carlsbad, CA, USA). Single cells were subsequently stained with Aldefluor (Stemcell technologies, Vancouver, BC, Canada) according to the manufacturer’s protocol. Cells were dissolved in Aldefluor assay buffer in the presence of activated Aldefluor. After 45 min, ALDH bright CASCs were flow-sorted using a FACSAria (Beckton&Dickinson, Franklin Lakes, NJ, USA).

The characteristics of the patients used to isolate and characterize CASCs are shown in Table 2.

### 4.2. Cell Culture

Isolated CASCs were cultured in X-vivo 15 medium (Lonza, Basel, Switzerland) supplemented with 2% penicillin/streptomycin (PS; Lonza, Basel, Switzerland) and 10% fetal bovine serum (FBS; GE Healthcare HyClone, Chicago, IL, USA). 293T and SW480 cell lines were cultured in Dulbecco’s modified Eagle medium/Nutrient Mixture F-12 (DMEM-F12, GIBCO, Thermo Fisher Scientific, Waltham, MA, USA) medium (Life Technologies, Carlsbad, CA, USA) supplemented with 2% PS and 10% FBS. All cell cultures were expanded at 37 °C in a humidified atmosphere containing 5% CO_2_.

The immunophenotype of CASCs was continuously monitored at different passages throughout the culture period and remained stable without any specific alteration. Expanded CASCs remained CD34-, CD45-, and CD117-negative. In contrast, the cells stably expressed CD29, CD55^dim^, CD73, CD90, and CD105. For CD90 in particular, two subpopulations were demonstrated to be present, CD90^dim^ and CD90^+^ [19].

### 4.3. Flow Cytometric Analyses of ALDH^br^ Cells in Different Compartments of the Heart

Equal amounts of tissue (3 g) were taken from left and right atrial appendage (LAA and RAA), left and right atrium, left and right ventricle, septum, and apex of slaughterhouse pig hearts (*n* = 3). For human samples, LAA and RAA samples of the same patient were used (*n* = 33). In addition to the routinely isolated RAA, the LAA was obtained as a precaution to avoid thrombus formation in patients at risk of developing atrial fibrillation. Subsequently the percentage of ALDH^br^ cells was determined by Aldefluor staining as previously described and expressed as the percentage of ALDH^br^ cells with respect to total cells.

### 4.4. Drugs and Treatment

The following small-molecule inhibitors and stimulators of the Wnt signaling pathway, all dissolved in DMSO, were used at the indicated final concentrations: CHIR99021 (6 µM; Stemcell technologies; 72054, Vancouver, BC, Canada), IWP2 (4 and 20 µM; Sigma Aldrich; I0536, St. Louis, MI, USA), C59 (1 and 5 µM; Abcam; ab142216, Cambridge, UK), XAV939 (2 and 10 µM; Sigma Aldrich; X3004, St. Louis, MI, USA), and IWR1-endo (4 and 20 µM; Stemcell technologies; 72562, Vancouver, BC, Canada). For all Wnt-related experiments, except the serial clonogenic assays, low-glucose DMEM (Life Technologies, Carlsbad, CA, USA) with 2% PS and 2% FBS was used. Equal concentrations of DMSO were used as control. Cells were incubated for 48 h to the various treatments. For CHIR 99021, a prolonged incubation period was also tested (72 h, 96 h, and 120 h), but this resulted in a clear decrease in cell viability.

For all Wnt inhibitors (XAV939, IWR1-endo, IWP-2, and C59), a five-fold higher concentration was tested on SW480 cells, but this did not result in a more pronounced effect on total and β-catenin expression. Furthermore, a two-fold higher concentration of the Wnt activator CHIR 99021 was tested for CASC viability, but this led to cell death due to high toxicity.

### 4.5. RNA Isolation and Complementary DNA (cDNA) Synthesis

Total RNA was extracted using the RNeasy mini kit (Qiagen, Venlo, The Netherlands) and equal amounts of RNA were reverse-transcribed with the SuperScript III Reverse Transcriptase Kit (Invitrogen, Carlsbad, CA, USA) primed with a random hexamer primer according to the manufacturer’s instructions.

### 4.6. Conventional PCR

PCR reactions were performed on a C1000 Touch thermal cycler (Bio-RAD, Hercules, CA, USA). The final reaction master mix (26 µL) was composed of 1× PCR buffer, 2.5 mM MgCl_2_, 250 µM dNTPs, 50 mU/µL Taq DNA Polymerase (all from Applied Biosystems, Waltham, MA, USA), 500 nM primer (Eurogentec, Seraing, Belgium), and 1.5 µL of cDNA. Primers sequences with corresponding annealing temperature and expected fragment size are listed in Table 3. Primers were designed to flank a large intron or span an exon–exon junction, when possible, to exclude genomic DNA contamination. Gene expression was visualized by gel electrophoresis. *β-actin* was used as an internal control.

### 4.7. Quantitative PCR

Real-time PCR reactions were carried out in duplicate with the Rotor-Gene Q (Qiagen, Venlo, The Netherlands) according to the MIQE guidelines [36]. The final reaction master mix composition was as follows: 1× Platinum^®^ SYBR^®^ Green qPCR SuperMix-UDG (Invitrogen, Carlsbad, CA, USA), 2 mM extra MgCl2 (only for *TBX5* and *TBX18*; Invitrogen, Carlsbad, CA, USA), and 2 µL of cDNA diluted 10-fold with DNase/RNase-free water. Primer sequences with corresponding annealing temperature and concentration are listed in Table 4. Primers were designed to flank a large intron or span an exon–exon junction, when possible, to exclude genomic DNA contamination. The PCR conditions were as follows: 50 °C for 2 min, 95 °C for 2 min, 40 cycles of 95 °C for 15 s, and annealing temperature for 30 s. For myosin light chain 7 (*MYL7*), an additional step of 80 °C for 30 s was added to each cycle. SW480 cells were used as a positive control for Brachyury transcription factor (*T*). The most stable reference genes for each experimental set-up were identified with NormFinder (Version 0.953, available at http://moma.dk/normfinder-software).

### 4.8. Western Blot Analysis

For total cellular protein, cells were immediately lysed with NP-40 lysis buffer (1% NP-40, 20 mM Tris-HCl, 137 mM NaCl, 10% glycerol, 2 mM EDTA) supplemented with Halt Protease inhibitor cocktail (1X; Life Technologies, Carlsbad, CA, USA) and 1 mM sodium orthovanadate. Equal amounts of protein, as measured by NanoDrop (Thermo Scientific, Waltham, MA, USA), were electrophoresed on 4–15% Mini-PROTEAN TGX gels and transferred to 0.2-µm nitrocellulose membranes using the Trans-Blot Turbo Transfer System (all from Bio-Rad, Hercules, CA, USA). Blots were blocked in Odyssey blocking buffer (Li-COR, Lincoln, NE, USA) for 1 h and probed with primary and secondary antibodies for 2 h and 45 min, respectively. The primary antibodies used were rabbit anti-β-catenin (D1048) mAb (1:1000; Cell signaling Technology; 8480; Danvers, MA, USA), mouse anti-active β-catenin mAb clone 8E7 (1:1000; Millipore; 05-665; Burlington, MA, USA), mouse anti-glyceraldehyde 3-phosphate dehydrogenase (GAPDH) mAb (1:2000; Abcam; ab8245; Cambridge, UK), and rabbit anti-GAPDH mAb (1:5000; Abcam; ab128915; Cambridge, UK). The following infrared fluorescently labeled secondary antibodies were used: goat anti-mouse IRDye^®^ 800 CW (1:15000; LI-COR; 926-32210, Lincoln, NE, USA) and goat anti-rabbit IRDye^®^ 680 RD (1:15000; LI-COR; 926-68071, Lincoln, NE, USA). Detection and quantification were accomplished using the Odyssey Infrared Imaging System and the LI-COR Odyssey Imaging software 2.1 (Lincoln, NE, USA). Data are shown as ratios normalized to the loading control GAPDH and compared with their respective control condition.

### 4.9. Viability Testing

CASC viability was measured with the fluorescein isothiocyanate (FITC) annexin V Apoptosis Detection Kit II (BD Pharmingen, Franklin Lakes, NJ, USA) according to the manufacturer’s manual. Early apoptotic cells were defined as annexin V-positive and propidium iodide (PI)-negative, while cells positive for both annexin V and PI were defined as late apoptotic cells. Viable cells were negative for both annexin V and PI. The percentage of viable cells was measured by a FACSAria and expressed as the percentage annexin V- and PI-negative cells of total cells.

### 4.10. Serial Clonogenic Assay

The clonogenic and proliferative potentials of CASCs were assessed by serial cloning. CASCs were cultured in X-vivo 15 medium with 2% PS and 20% FBS, supplemented with or without 6 µM CHIR99021 for the treatment and control condition, respectively. In the treatment group, continuous exposure to CHIR 99021 was used.

Prior to assessing the clonogenic character, CASCs were labeled with green fluorescent protein (GFP). Production of GFP-containing lentiviruses and transduction of CASCs was performed as previously described [14]. In brief, for viral production, 293T cells were transfected with the cytomegalovirus GFP mammalian expression plasmid (pRRL-CMV-GFP) together with the packaging plasmids pMDLg-RRE, pRSV-REV, and pCMV-VSVG, using EZLentifect (MellGen Laboratories, San Diego, CA, USA). For GFP labeling, CASCS were incubated with viral supernatant containing replication-defective lentiviruses supplemented with 8 pg/mL polybrene (Sigma-Aldrich, St. Louis, MI, USA). Subsequently, GFP-positive CASCs were flow-sorted with FACSAria in 96-well plates at a density of one cell per well. Single-cell disposition was microscopically analyzed, and wells with no cells or more than one cell were excluded. Medium exchange and scoring of the number and size of colonies was performed every four days. Clonogenicity (%) was defined as the number of wells with >1 cell, >2 cells, >5 cells, >10 cells, or >50 cells to the total number of wells. At days 8 and 12, clones (> 50 cells) were harvested and reseeded at a density of one cell per well to generate secondary clones. Recloning of individual clones continued until cloning activity was exhausted.

### 4.11. Statistics

All quantitative results are presented as medians ± IQR. When d’Agostino and Pearson testing revealed that the data were not normally distributed, comparisons between two paired groups were performed in GraphPad version 6.01 (GraphPad Software Inc, San Diego, CA, USA) using the non-parametric Wilcoxon signed rank test. To compare more than two paired groups, the non-parametric Friedman test with Dunn’s post hoc test was used. For the comparison of the percentage of ALDH^br^ cells in LAA and RAA of patients, a paired parametric *t*-test was used. A value of *p* < 0.05 was considered significant.

## 5. Conclusions

In conclusion, these results indicate that CASCs are predominantly present in the atrial appendages, are committed toward myocardial differentiation, and express several FZD receptors. However, despite the crucial role of Wnt signaling in cardiogenesis and differentiation of PSCs towards cardiac lineages, Wnt signaling is functional but has limited effects on CASCs. Further research is, therefore, necessary to identify the molecular mechanisms that are involved in CASC proliferation and differentiation.

## 6. Patents

The technology for isolating CASCs is patented: EP 2 258 833 B1

## Figures and Tables

**Figure 1 ijms-21-03931-f001:**
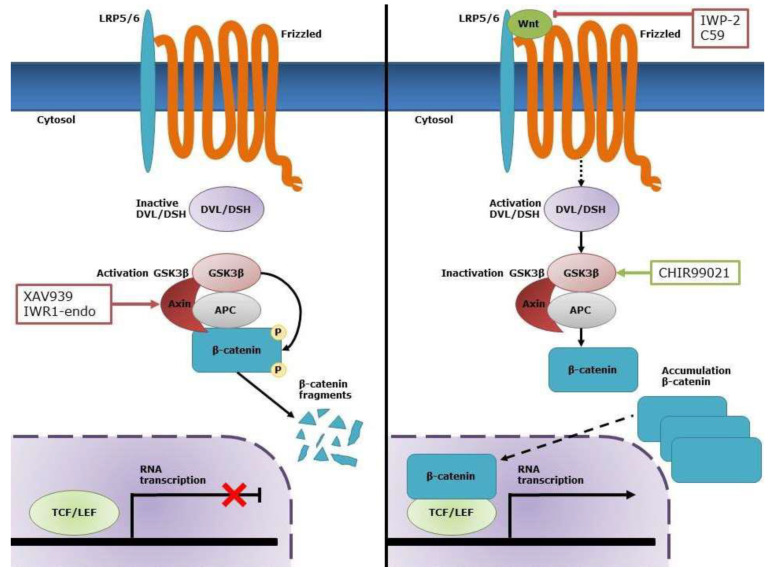
Overview of the canonical Wnt/β-catenin pathway. In the absence of Wnt, β-catenin is phosphorylated by the destruction complex consisting of Axin, adenomatous polyposis coli (APC), and glycogen synthase kinase (GSK)-3β, which leads to its degradation. In the presence of Wnt, this ligand interacts with its receptor complex consisting of low-density lipoprotein receptor-related protein 5/6 (LRP5/6) and Frizzled (FZD). This results in the activation of Disheveled (DSH) and subsequent inactivation of GSK-3β. As a result, β-catenin accumulates in the cytosol and translocates to the nucleus where it activates transcription of specific target genes through interaction with the T-cell factor/lymphoid enhancer factor (TCF/LEF) transcription factors. CHIR99021 is a selective inhibitor of GSK-3β and, therefore, increases the cytosolic levels of β-catenin, which allows nuclear translocation of active β-catenin. C59 and IWP2 are both inhibitors for Wnt production and secretion, while XAV939 and IWR1-endo inhibit Wnt signaling via stabilization of the APC/Axin/GSK-3β destruction complex of β-catenin.

**Figure 2 ijms-21-03931-f002:**
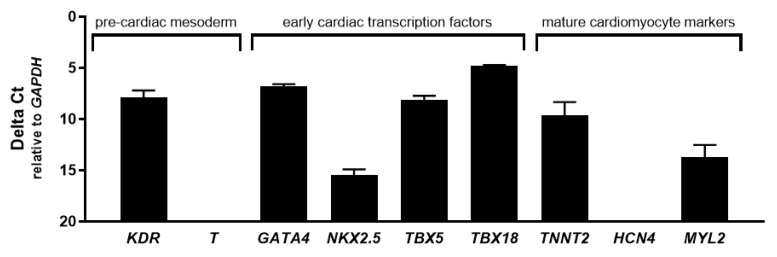
Early but no late cardiac differentiation markers are expressed in human cardiac atrial appendage stem cells (CASCs) during culture. Messenger RNA (mRNA) expression of *KDR* and *T* (pre-cardiac mesoderm); *GATA4*, *NKX2.5*, *TBX5*, and *TBX18* (early cardiac transcription factors); *TNNT2*, *HCN4*, *MYL2*, and *MYL7* (mature cardiomyocyte markers). Data are shown as medians ± interquartile range (IQR) (*n* = 3 for individual patient CASC cultures).

**Figure 3 ijms-21-03931-f003:**
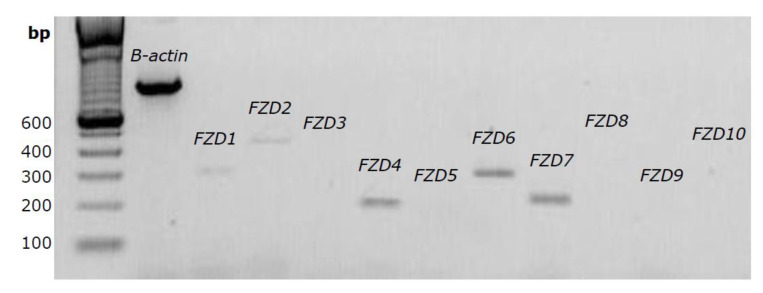
Several FZD receptors are expressed in human CASCs. Representative gel of *FZD1* to *FZD10* expression after 25 PCR cycles. *β-actin* was used as internal control.

**Figure 4 ijms-21-03931-f004:**
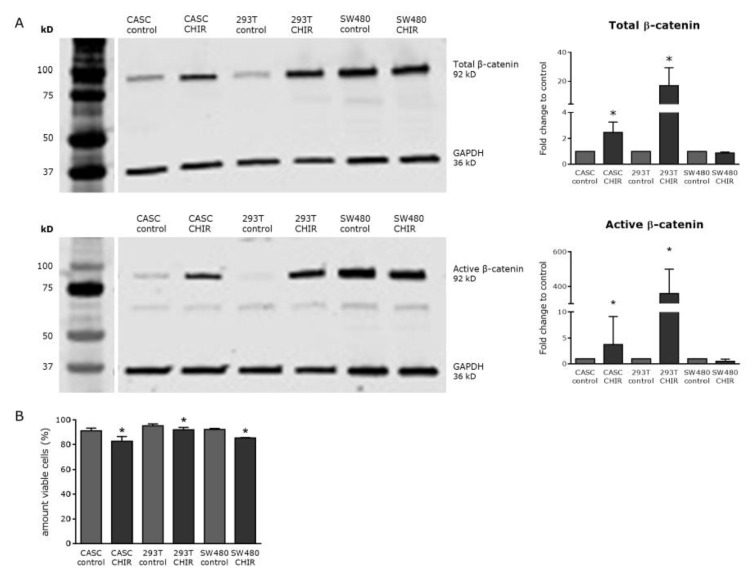
CHIR99021 is a potent Wnt activator in CASCs but slightly decreased its viability. (**A**) Representative Western blots (left panels) and subsequent quantification (right panels) of both total and active β-catenin after CHIR99021 treatment. (**B**) Cell viability of CASCs, as well as 293T and SW480 cells, treated with 6 µM CHIR99021. Data are shown as medians ± IQR (*n* = 6 individual patient CASC cultures/condition); * *p* < 0.05 compared to respective control.

**Figure 5 ijms-21-03931-f005:**
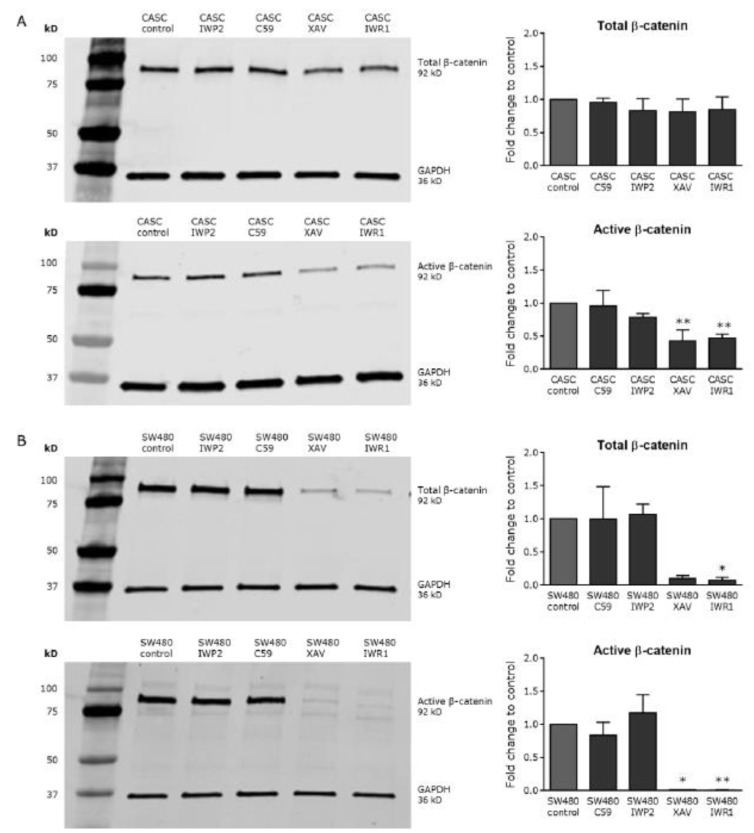
XAV939 and IWR1-endo decrease Wnt activity in CASCs. Representative Western blots (left panels) and subsequent quantification (right panels) of both total and active β-catenin after treatment with small-molecule Wnt inhibitors in (**A**) CASCs and (**B**) SW480 cells. Data bars represent medians ± IQR (*n* = 6 individual patient CASC cultures/condition); * *p* < 0.05, ** *p* < 0.01 compared to respective control.

**Figure 6 ijms-21-03931-f006:**
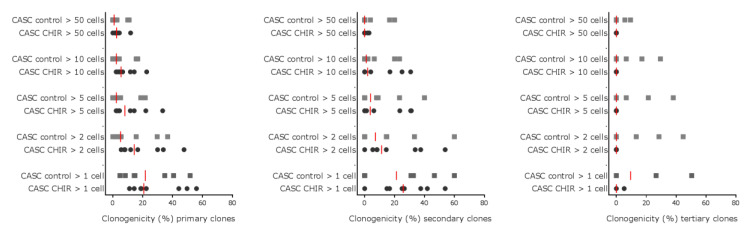
Wnt signaling does not influence CASC clonogenicity or proliferation. The clonogenic potential of CASCs was studied using a serial cloning assay. Individual data points are shown with the red line representing the median of each condition (*n* = 8 individual patient CASC cultures/condition).

**Figure 7 ijms-21-03931-f007:**
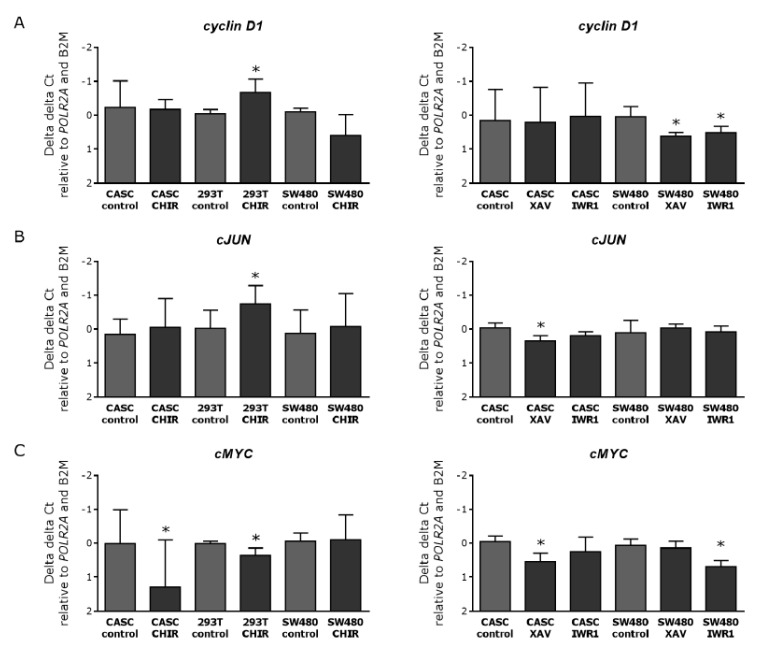
Wnt stimulation and inhibition in CASCs have no clear effect on the expression of proliferation-associated genes. mRNA expression of the Wnt proliferation-associated target genes (**A**) *Cyclin D1*, (**B**) *cJUN*, and (**C**) *cMYC* after Wnt stimulation (left panels) and inhibition (right panels) in CASCs. Data are shown as medians ± IQR (*n* = 6 individual patient CASC cultures/condition); * *p* < 0.05 compared to respective control.

**Figure 8 ijms-21-03931-f008:**
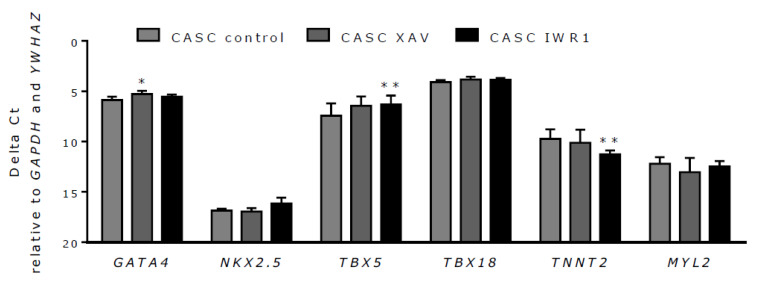
Wnt inhibition has a limited effect on CASC differentiation. mRNA expression levels of the cardiac differentiation markers *GATA4*, *NKX2.5*, *TBX5*, *TBX18*, *TNNT2*, and *MYL2* after treatment with XAV939 and IWR1-endo. Data are shown as medians ± IQR (*n* = 6 individual patient CASC cultures/condition); * *p* < 0.05, ** *p* < 0.01 compared to respective control.

**Table 1 ijms-21-03931-t001:** Percentages of aldehyde dehydrogenase bright (ALDH^br^) cells in different compartments of the pig heart.

	ALDH^br^ Cells (%)		
	Pig heart 1	Pig heart 2	Pig heart 3
LAA	4.4	4.9	3.5
RAA	10.5	5.6	8.7
Left atrium	0.4	1.5	1.5
Right atrium	2.8	1.5	3.9
Left-ventricular free wall	0.5	0.4	0.2
Right-ventricular free wall	1.1	1.9	1.9
Apex	3.1	1.6	0.2
Septum	0.3	0.4	0.2

LAA = left atrial appendage; RAA = right atrial appendage. Data are as individual datapoints (*n* = 3).

**Table 2 ijms-21-03931-t002:** Patient characteristics.

Patient Characteristics	
Age (years)	70 ± 14 ^a^
Gender: male/female (%)	70/30 ^a^
Risk factors	
Weight (kg)	80 ± 15 ^a^
Body mass index (kg/m²)	28 ± 5 ^a^
Last creatinine level pre-operation (mg/dL)	1.0 ± 0.3 ^a^
Smoking: current smoker/ex-smoker ≥1 month/never smoked (%)	15/36/48 ^a^
Diabetes (%)	33 ^a^
Hypercholesterolemia (%)	87 ^c^
Renal dysfunction (%)	9 ^b^
Hypertension (%)	61 ^c^
Chronic lung disease (%)	16 ^b^
Peripheral vascular disease (%)	13 ^b^
Cerebrovascular disease (%)	19 ^b^
Pre-operative cardiac status	
Myocardial infarction (%)	34 ^b^
Congestive heart failure (%)	16 ^c^
Angina: CCS 0/I/II/III/IV (%)	18/18/57/4/4 ^d^
Classification: NYHA I/II/III/IV (%)	91/18/14/0 ^d^
Surgical procedure	
Number of vessels: 0/1/2/3 (%)	9/3/27/61 ^a^
CABG/valve/other (%)	91/15/3 ^a, e^

CABG = Coronary Artery Bypass Grafting; CCS = Canadian Cardiovascular Society; NYHA = New York Heart Association. Values are expressed as medians ± IQR or as percentages of the total patient population. ^a^
*n* = 33, ^b^
*n* = 32, ^c^
*n* = 31, ^d^
*n* = 28; ^e^ 3 patients underwent a combined procedure.

**Table 3 ijms-21-03931-t003:** Primer sequence, annealing temperature, and fragment size.

Primer	Primer Sequence	Annealing Temperature	Fragment Size
*β-actin*	Forward: 5′–AGCGGGAAATCGTGCGTGACA–3′ Reverse: 5′–CCTGTAACAATGCATCTCATATTTGG–3′	56 °C	791 bp
*FZD1*	Forward: 5′–CCTTTCTTTCCTGGCTTGA–3′ Reverse: 5′–CTCACCCTGTAACCAACTAAG–3′	56 °C	285 bp
*FZD2*	Forward: 5′–CCCTACTCATTTGTCCTGTC–3′ Reverse: 5′–TGAATAGACTGCAGGGAAAG–3′	56 °C	405 bp
*FZD3*	Forward: 5′–CTCCTGAGGGATCCAAATAC–3′ Reverse: 5′–GAGCCGATGAGAACTACTATG–3′	56 °C	282 bp
*FZD4*	Forward: 5′–GAGAGAGAAGAGAGGAAATGG–3′ Reverse: 5′–GGTCACTTAATTGTTGCTAGTT–3′	56 °C	185 bp
*FZD5*	Forward: 5′–CCCAGAGCTAGGAAATGTAG–3′ Reverse: 5′–GATGTGCTCTGTCCTGTT–3′	56 °C	165 bp
*FZD6*	Forward: 5′–CATCAATGAGAGAGGTGAAAG–3′ Reverse: 5′–GGGTGAACAAGCAGAGAT–3′	56 °C	286 bp
*FZD7*	Forward: 5′–CATTTGGATCCTTTGAGGTAAA–3′ Reverse: 5′–CTCTTCGTTCACTATGGTATCT–3′	56 °C	203 bp
*FZD8*	Forward: 5′–CTCTTCCTACGTAAACTCCC–3′ Reverse: 5′–GAGAGGGCAATGGTTAAATC–3′	56 °C	356 bp
*FZD9*	Forward: 5′–CTCCAAGACTTTCCAGACC–3′ Reverse: 5′–GTCCGTCTTAGTCATGTGC–3′	56 °C	157 bp
*FZD10*	Forward: 5′–GATTCAGCCCTCAGAAGAA–3′ Reverse: 5′–GCAGAGAGACTATTGGTGAA–3′	56 °C	292 bp

*FZD* = Frizzled receptor.

**Table 4 ijms-21-03931-t004:** Primer sequence, annealing temperature and concentration.

Primer	Primer Sequence	Annealing Temperature	Concentration
*B2M*	Forward: 5′–AAGATGAGTATGCCTGCCGT–3′ Reverse: 5′–TTCATCCAATCCAAATGCGGC–3′	60 °C	300 nM
*cJUN*	Forward: 5′–AGGTGGAGTTGAAAGAGTTAAGA–3′ Reverse: 5′–ACCATAGCATCAGGTACATCAG–3′	60 °C	300 nM
*cMYC*	Forward: 5′–TTCTCTGAAAGGCTCTCCT–3′ Reverse: 5′–GTGAAGCTAACGTTGAGGG–3′	60 °C	300 nM
*Cyclin D1*	Forward: 5′–TCAAATGTGTGCAGAAGGAG–3′ Reverse: 5′–TCTCCTTCATCTTAGAGGCC–3′	60 °C	100 nM
*GAPDH*	Forward: 5′–AGTCAACGGATTTGGTCGTATTG–3′ Reverse: 5′–ATCTCGCTCCTGGAAGATGGT–3′	60 °C	300 nM
*GATA4*	Forward: 5′–ACCTGAATAAATCTAAGACACCAG–3′ Reverse: 5′–CATCGCACTGACTGAGAAC–3′	60 °C	100 nM
*HCN4*	Forward: 5′–AGTCGGCCGGATTTTGGATT–3′ Reverse: 5′–AGGTGATGCCCACAGGAATG–3′	58 °C	100 nM
*KDR*	Forward: 5′–CTAGGTAAGCCTCTTGGCCG–3′ Reverse: 5′–CGATGCTCACTGTGTGTTGC–3′	66 °C	300 nM
*MYL2*	Forward: 5′–TTGGGCGAGTGAACGTGAAA–3′ Reverse: 5′–GGTCCGCTCCCTTAAGTTTCT–3′	60 °C	300 nM
*MYL7*	Forward: 5′–GGAGTTCAAAGAAGCCTTCAGC–3′ Reverse: 5′–GTCAGGGCGAACATCTGCT–3′	60 °C	300 nM
*NKX2.5*	Forward: 5′–CAAGGACCCTAGAGCCGAAA–3′ Reverse: 5′–CACCGACACGTCTCACTCAG–3′	58 °C	450 nM
*POLR2A*	Forward: 5′–TCACAGCAGTGCGCAAATTC–3′ Reverse: 5′–CCACGTCGACAGGAACATCA–3′	60 °C	300 nM
*T*	Forward: 5′–ACTCCCAATCCTATTCTGACAACT–3′ Reverse: 5′–CGTTGCTCACAGACCACAGG–3′	60 °C	450 nM
*TBX5*	Forward: 5′–CAGGAGCATAGCCAAATTTACCA–3′ Reverse: 5′–GGATAGCTAGAGCGGTAGAAGGA–3′	62 °C	300 nM
*TBX18*	Forward: 5′–ATGCATTCTGGCGACCATCA–3′ Reverse: 5′–ACGCCATTCCCAGTACCTTG–3′	62 °C	300 nM
*TNNT2*	Forward: 5′–ACTTGGAGGCAGAGAAGTTCG–3′ Reverse: 5′–CGGTGACTTTAGCCTTCCCG–3′	66 °C	100 nM
*YHWAZ*	Forward: 5′–AGACGGAAGGTGCTGAGAAAA–3′ Reverse: 5′–TGTGAAGCATTGGGGATCAAGA–3′	60 °C	300 nM

*B2M* = Betβa-2-Microglobulin; *cJUN* = Jun proto-oncogene; *cMYC* = V-Myc avian myelocytomatosis viral oncogene homolog; *GAPDH* = glyceraldehyde 3-phosphate dehydrogenase; *GATA4* = GATA binding protein 4; *HCN4* = hyperpolarization-activated cyclic nucleotide-gated potassium channel 4; *KDR* = kinase insert domain receptor; *MYL2* = myosin light chain 2; *MYL7* = myosin light chain 7; *NKX2.5* = NK2 homeobox 5; *POLR2A* = polymerase (RNA) II subunit A; *T* = T Brachyury transcription factor*; TBX5* = T-box 5; *TBX18* = T-box 18; *TNNT2* = troponin T2, cardiac type; *YWHAZ* = tyrosine 3-monooxygenase/tryptophan 5-monooxygenase activation protein zeta.

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
