# Peer review of "Differentiation of Human Cardiac Atrial Appendage Stem Cells into Adult Cardiomyocytes: A Role for the Wnt Pathway?"

_ijms, 2020, doi:10.3390/ijms21113931_

Round 1
Reviewer 1 Report
The manuscript by Willems et al. intends to investigate a possible effect of the WNT pathway on clonogenicity or cardiac maturation of Cardiac stem cell derived from right atrial appendages. Initially they demonstrate how the right atrial appendage is the best resource of this cellular type. Subsequently they make a molecular characterization, proving the feasibility of the WNT modulation. After WNT activation or inhibition the authors control the differentiation or proliferative effects concluding that the modulation of the WNT pathway alone is not sufficient to induce the investigated effects. However, the present study is very preliminary and need some major revisions.
- It is known that the timing and doses of activators and inhibitors of different pathways play a fundamental role in cardiac commitment. Why do the authors choose the specific doses reported? Although used in other cell types, CASCs may respond differently. It may be necessary to evaluate the different dose curves.
- The authors do not report the duration of the treatments. Long incubations (48-72h) may be required for changes in gene expression. This aspect may also need to be assessed.
- After how long from treatment are gene expression and β-catenin activation or inhibition analyzed? The expression it may take longer to change. Analyze the expression effects at different time points.
- It is not clear to me whether the inhibition of the pathway is preceded by its activation via CHIR. If so, what are the timelines adopted? if not, how do the authors explain a commitment effect of these inhibitors since the pathway is not, or very weakly, activated in control condition (figure 4). It would be necessary to evaluate a multi steps protocol to achieve differentiation.
- The choice of mature cardiomyocyte markers is questionable. The authors have already published that CASC expressed many sarcomeric genes at the mRNA level, however protein expression was achieved only after co-culture. Why hasn't protein expression and sarcomeric organization been investigated by immunofluorescence? Why not verify the expression of more functional markers already reported for these CASC cells such as the expression of MYH6, connexin43 and ion channels (kv4.3 / CACNA1C). The reported HCN4 channels is a specific marker of sinus node, and links to spontaneous beating activity rarely reported for CASC even after long co-colture. If the authors want to link the trascriptional factors to a sinus commitment please explain why, if not change the markers of interest.
- Comparing expression data to a single atrium and ventricle sample has no meaning, increase the number or remove this type of comparison.
- The percentage of CASC reported in this paper are much higher than those previously published for the same tissue district. Why? Please discuss this aspect.
- It would be useful to show representative FACS dot plots in which the positivity area is identified.
- Why did the authors choose non-parametric statistical tests? data is not normally distributed? if yes explicit, if this is not the case (as usually expected), carry out parametric tests.
Reviewer 2 Report
The authors evaluated cardiac atrial appendage stem cells (CASC) and ways to differentiate them toward cardiomyocytes. Through expression assay, they found that these cells resemble cardiac progenitors. The WNT signaling can be modulated in these cells by small molecules. Finally, the WNT inhibition itself wasn't sufficient to induce cardiomyocyte differentiation.
- Cardiac stem cells is a relatively controversial topic, given the recent news on the c-kit cells. The authors should indicate the difference/relationship between CASCs and c-kit cells.
- The identity of CASC is of great importance. A high throughput transcriptome profiling approach is essential for the elucidation.
- Do these cells contract in response to stimulation. Is it possible to record action potentials from them? Initial phenotyping is also warranted.
- How did the authors obtain the CASCs? A few flow cytometry plots should be present.
- Brachyury is usually considered to be marker of primitive streak. For cardiac mesoserm, MESP1, MEF2c, ISL1, etc. can be evaluated.
- Should also test the expression of pluripotency markers including OCT4, Nanog, etc. in untreated cells.
- For figure 3, is it conventional PCR or quantitative real time PCR?
- For figure 7, how to explain that both WNT activation and inhibition decreased the cMYC expression? Should also measure the cardiac mesoderm/cardiomyocyte gene expression here.
Round 2
Reviewer 1 Report
No other comments.
Author Response
Answer to Reviewers (round 2)
We thank the Reviewers for allowing us to further improve the quality of the manuscript, following one remaining remark. We hope that, with the answer provided and the modifications made to the manuscript with track changes, the manuscript is now acceptable for publication.
Reviewer 1
No further modifications were made to the manuscript, as the Reviewer had no further comments. We want to thank him/her for accepting our answers to his previous review.
Reviewer 2
The author has addressed most of my concerns. Just one comment. In Figure 3, regular PCR was performed to detect various FZD proteins. While it is fine to detect the presence of individual protein by this approach, it is not a good idea to do quantitative analysis in this context, one of reasons being that the density of gel band varies with position. I would recommend using quantitative fluorescent PCR for improvement if necessary.
We acknowledge this comment and agree with the Reviewer that for quantification, quantitative fluorescent PCR would be advisable. However, it was not our intention to quantify the expression of the genes (and we apologize if we wrongly gave that impression) but rather to demonstrate the presence of the FZD receptor, and to show which receptor subtypes are expressed.
Therefore, we removed the right panel in Fig. 3 and modified the Fig legend accordingly (p.6, l.13-15):
Figure 3. Several FZD receptors are expressed in human CASCs. Representative gel of FZD1 to FZD10 expression after 25 PCR cycles. β-actin was used as internal control.
In the materials and methods section (p. 15 l.44): “For quantification, densitometry was performed with ImageJ software and normalized to ß-actin” was removed.
Reviewer 2 Report
The author has addressed most of my concerns. Just one comment. In Figure 3, regular PCR was performed to detect various FZD proteins. While it is fine to detect the presence of individual protein by this approach, it is not a good idea to do quantitative analysis in this context, one of reasons being that the density of gel band varies with position. I would recommend using quantitative fluorescent PCR for improvement if necessary.
Author Response

(The authors gave the same response as above.)
